# Theta oscillations optimize a speed-precision trade-off in phase coding neurons

**Adrián F. Amil** [1]ᵒ*, **Albert Albesa-González**[2]ᵒ, **Paul F. M. J. Verschure**[3,4]

1 Donders Institute for Brain, Cognition and Behaviour–Radboud Universiteit, Nijmegen, The Netherlands, 2 Department of Bioengineering, Imperial College London, London, United Kingdom, 3 Instituto de Neurociencias de Alicante, Consejo Superior de Investigaciones Científicas (CSIC)–Universidad Miguel Hernández de Elche, Alicante, Spain, 4 Department of Health Psychology, Universidad Miguel Hernández de Elche, Alicante, Spain

ᵒ These authors contributed equally to this work.
* adrian.fernandezamil@donders.ru.nl

**Data Availability Statement:** The code and data to reproduce all figures can be found at https://github.com/adriamilcar/Phase-Coding-in-LIF.

## Abstract

Theta-band oscillations (3–8 Hz) in the mammalian hippocampus organize the temporal structure of cortical inputs, resulting in a phase code that enables rhythmic input sampling for episodic memory formation and spatial navigation. However, it remains unclear what evolutionary pressures might have driven the selection of theta over higher-frequency bands that could potentially provide increased input sampling resolution. Here, we address this question by introducing a theoretical framework that combines the efficient coding and neural oscillatory sampling hypotheses, focusing on the information rate (bits/s) of phase coding neurons. We demonstrate that physiologically realistic noise levels create a trade-off between the speed of input sampling, determined by oscillation frequency, and encoding precision in rodent hippocampal neurons. This speed-precision trade-off results in a maximum information rate of ∼1–2 bits/s within the theta frequency band, thus confining the optimal oscillation frequency to the low end of the spectrum. We also show that this framework accounts for key hippocampal features, such as the preservation of the theta band along the dorsoventral axis despite physiological gradients, and the modulation of theta frequency and amplitude by running speed. Extending the analysis beyond the hippocampus, we propose that theta oscillations could also support efficient stimulus encoding in the visual cortex and olfactory bulb. More broadly, our framework lays the foundation for studying how system features, such as noise, constrain the optimal sampling frequencies in both biological and artificial brains.

## Author summary

The mammalian hippocampus exhibits prominent oscillations in the theta band (3–8 Hz) during exploration, enabling individual neurons to rhythmically sample and represent input signals from the cortex. However, the reason behind the specific frequency of this hippocampal rhythm has remained unclear. In this study, we developed a biologically-based theoretical framework to demonstrate that neurons using oscillations to efficiently

**Funding:** This work was supported by a HORIZON-EIC-2021 Pathfinder Challenges grant to CAVAA, project number 101071178 (to P.F.M.J.V.). The funders had no role in study design, data collection and analysis, decision to publish, or preparation of the manuscript.

**Competing interests:** The authors have declared that no competing interests exist.

sample noisy signals encounter a trade-off between their sampling speed (i.e., oscillation frequency) and their coding precision (i.e., reliability of encoding). Notably, our findings reveal that this trade-off is optimized precisely within the theta band, while also providing insights into other fundamental features of the hippocampus. In conclusion, we offer an explanation grounded in efficient coding for why hippocampal oscillations are confined to the theta band and establish a foundation for exploring how the properties of neurons determine optimal sampling frequencies across neural circuits.

## Introduction

Early physiological experiments with rodents revealed that certain neurons exhibit increased firing rates when the animal occupies specific regions within its environment, referred to as place fields [1]. Further research revealed that these place cells also exhibit a temporal pattern known as phase precession: as the animal traverses a place field, they progressively fire at earlier phases of the local field potential (LFP) oscillations [2]. This discovery spurred extensive research into the relationship between behavior and the timing of neuronal firing relative to theta-band (3–8 Hz) oscillations. It was found that incorporating firing phase information alongside firing rates significantly enhanced the accuracy of reconstructing the animal's position [3], establishing the phase code as a viable alternative to the traditional firing rate code in neural circuits [4].

The phase precession phenomenon also led to the development of theories and models that would explain the temporal progression of spikes across the theta phase. One influential model is the rate-to-phase transform, where firing rate input from the entorhinal cortex make hippocampal neurons to fire earlier or later depending on firing rate levels, transforming rate-coded inputs into phase-coded outputs [5, 6]. Empirical evidence supporting this model [7, 8] has also been observed outside the hippocampus [9], suggesting a broader application to sensory-related areas [4]. Indeed, the rate-to-phase transform can be linked to synaptic sampling [10], where low-frequency field oscillations modulate neuronal excitability–via ephaptic effects [11] or indirectly via feedback inhibition [12, 13]. By leveraging a rhythmically-organized first-spike latency code [14], phase coding can sample rate-based synaptic inputs each cycle. This rhythmic neural sampling from sensory-related areas would then manifest as perceptual sampling effects at the behavioral level [15–17]. Behavioral and electrophysiological experiments support this hypothesis, showing that stimulus detection is influenced by the timing of stimulus presentation relative to the oscillatory cycle in visual [18] and olfactory [19, 20] domains. Furthermore, the idea of rhythmic neural sampling aligns with the short membrane time constants of pyramidal cells (10–30 ms) compared to the periods of low-frequency field oscillations (100 ms to 1 s), allowing for cycle-to-cycle resets. Therefore, rhythmic field activity along the sensory hierarchy could provide a scaffold for representing sensory signals through repeated synaptic input sampling, suggesting that rhythmic sampling might also facilitate probabilistic inference of the underlying sensory streams [10, 16].

Following the efficient coding hypothesis [21], oscillations in brain circuits may have been evolutionarily optimized for information encoding and transmission. The crucial role of oscillations in neuronal communication [13], and the conservation of oscillatory rhythms [22] together with the observation of phase coding across mammalian species [2, 9, 23–25], seem to support this hypothesis. However, a fundamental question remains: why has this low-frequency range emerged over others that could offer increased bandwidth and sampling resolution? In this paper, we address why oscillations supporting phase coding seem to be confined

to low frequencies like the theta band. Through theoretical analysis and simulations, we show that theta-band frequencies strike a balance between sampling speed (i.e., oscillation frequency) and encoding precision (i.e., information per oscillation cycle), resulting in an optimal information rate (bits/s). We further extend our analysis to account for the persistence of theta oscillations along the hippocampal dorsoventral axis, their modulation by the animal's running speed, and also their presence in extra-hippocampal brain areas like the primary visual cortex and the olfactory bulb. Finally, we discuss how our approach can contribute towards understanding oscillations through the lenses of optimality and efficient coding.

## Materials and methods

We first introduce our theoretical framework, where we derive and validate an analytical approximation of the information rate conveyed by phase coding neurons.

### Neuron model

We model pyramidal cells as stochastic leaky integrate-and-fire (LIF) neurons receiving oscillatory and tonic inputs [26] (Fig 1). Their membrane potential dynamics are defined by:

$$\tau_m \frac{dV}{dt} = -V + R_m I_s - R_m I_{osc} \cos(\omega t) + \tau_m \sigma_W(I_s)\xi(t), \tag{1}$$

with $V(t' + dt) = 0$ if $V(t') \geq V_{th}$. Here, $V$ is the membrane potential, $R_m$ the membrane resistance, $\tau_m$ the membrane time constant, $I_s$ the tonic input current that conveys the stimulus information to be encoded, $I_{osc}$ the oscillation amplitude, $\omega = 2\pi f$ the angular frequency, $V_{th}$ the spike threshold, $\sigma_W(I_s)$ the signal-dependent noise amplitude, and $\xi(t) \equiv dW(t)/dt$ the Wiener process derivative with $\xi(t) \sim \mathcal{N}(0, 1/\sqrt{dt})$.

**Noise.** We assume input currents in our neuron model have a synaptic origin (Fig 1A). A high-rate Poisson input with synaptic time constants $\tau_s \ll \tau_m$ leads to Gaussian white noise, $\xi(t)$, in Eq 1. Although biological neurons tend to display temporally-correlated noise (e.g., pink noise) [28–30], Gaussian white noise has remained a reasonable approximation (see [26, 27]), with the additional advantage of being more analytically tractable. Under the assumption of white noise, the standard deviation of the free membrane potential (i.e., the membrane

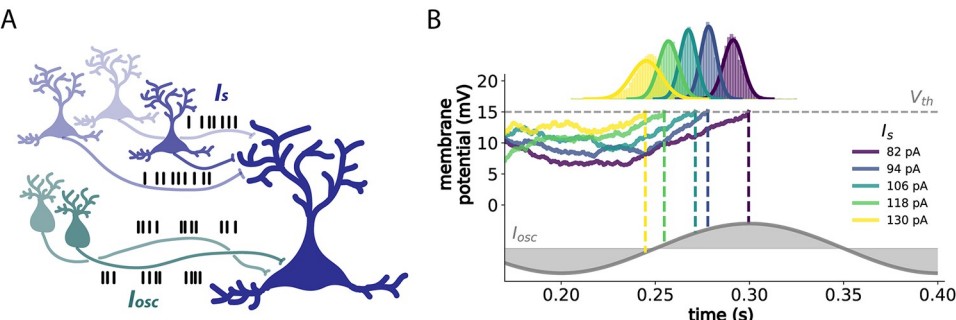

**Fig 1. Phase coding in LIF neurons.** (A) Model neuron receiving rhythmic volleys of inhibitory spikes to the soma, and continuous volleys of excitatory spikes to the dendrites, representing the oscillatory and tonic inputs in Eq 1, respectively. (B) Examples of voltage trajectories for such a neuron with hippocampal parameters [6] and realistic noise amplitudes [27] (see Table A in S4 Appendix for details on parameters), for a range of tonic inputs $I_s$ (color code). Vertical dashed lines denote the times at which the neuron reaches threshold (denoted by the horizontal dashed line), with respect to the oscillation (i.e., phase). Histograms and the corresponding Gaussian fits denote the probability distributions of spike times for each $I_s$.

potential without considering the threshold $V_{th}$), denoted as $\sigma_V$, is directly proportional to the noise strength $\sigma_W$. In turn, it can be demonstrated that $\sigma_V$ is proportional to $\sqrt{\bar{w}^2 \bar{v}}$, where $\bar{w}$ represents the mean synaptic efficacy, and $\bar{v}$ denotes the mean presynaptic firing rate [16]. Defining the effective tonic input as $I_s = \bar{w}\bar{v}$, it then follows that $\sigma_V \propto \sqrt{I_s^2/\bar{v}}$. Importantly, due to the membrane's low-pass filtering properties ($\tau_m$), $I_s$ must increase with oscillation frequency $f$ to maintain phase-locking (S1C and S1D Fig). We assume these adjustments result from changes in $\bar{w}$, keeping $\bar{v}$ constant across frequencies. Hence, we incorporate signal-dependent noise from Poisson processes (where variance equals the mean) by scaling the noise amplitude $\sigma_W$ by $KI_s$, where $K$ is a constant representing $1/I_{s,f=1}$, that is, the inverse of the weakest $I_s$ possible that makes the neuron phase-lock to a 1 Hz oscillation—acting as a baseline. Therefore, $\sigma_W$ can be defined as:

$$\sigma_W(I_s) = KI_s \eta \frac{V_{th}}{\sqrt{\tau_m}}, \tag{2}$$

with $KI_s$ as the signal-dependent factor akin to the linear modulation of noise amplitude in motor control [31, 32], $\eta$ a dimensionless parameter controlling noise strength, and $\frac{V_{th}}{\sqrt{\tau_m}}$ scaling $\sigma_W$ and giving it noise units $V/\sqrt{s}$.

## Approximation of spike phase distributions

Given the neuron model described by Eqs 1 and 2, we can proceed to derive an analytical approximation for the mean and variance of the spike phase distributions (e.g., Fig 1B).

**Mean phase.** As demonstrated in previous work [6], we can derive a closed-form expression for the expected phase $\mu_\phi$ at which the neuron described by Eq 1 will phase-lock. By integrating the deterministic part of Eq 1, we can find the phase at which the expected trajectory of the membrane potential $\mu_V(t)$ intersects with the spike threshold $V_{th}$, exactly after one oscillation period $T$ (see the S1 Appendix for the full derivation), obtaining:

$$\mu_\phi = \arccos\left(\frac{R_m I_s(1 - e^{-T/\tau_m}) - V_{th}}{R_m I_{osc} A(1 - e^{-T/\tau_m})}\right) - \varphi, \tag{3}$$

where $A = 1/\sqrt{1 + (\tau_m\omega)^2}$, representing the filtering of the oscillatory input by the membrane, and $\varphi = -\arctan(\omega\tau_m)$, representing a phase shift. Eq 3 captures how the integrated tonic and oscillatory inputs make a neuron reach threshold $V_{th}$ with a period $T$ between spikes (see S1A–S1C Fig for details).

**Phase variance.** To estimate the variance of spike phase $\sigma_\phi^2$ due to noise, we consider the variance in spike timing $\sigma_t^2$ induced by fluctuations in the membrane potential. Similarly to previous work on phase jitter in oscillators [33, 34], we employ a first-order Taylor approximation of the membrane dynamics around the spike threshold $V_{th}$, such that $\delta V \approx \frac{dV}{dt}\delta t$, and consider the integrated noise $\sigma_V^2(t = T) = \frac{K^2 I_s^2 \eta^2 V_{th}^2}{2}(1 - e^{-2T/\tau_m})$. Using the propagation of uncertainty, we express $\sigma_t^2 \approx \left(\frac{\delta t}{\delta V}\right)^2 \sigma_V^2$. Then, since time and phase are related by $\phi = \omega t$, we have that $\sigma_\phi^2 \approx \omega^2 \sigma_{t_f}^2$. Hence, we arrive at the following approximation (see Fig 2A, and S2 Appendix for more details):

$$\sigma_\phi^2 \approx \omega^2 \left.\frac{\sigma_V^2}{\left(\frac{dV}{dt}\right)^2}\right|_{V_{th}} = \frac{\omega^2 K^2 I_s^2 \eta^2 V_{th}^2(1 - e^{-2T/\tau_m})\tau_m^2}{2(-V_{th} + R_m I_s - R_m I_{osc}\cos(\mu_\phi))^2}. \tag{4}$$

Eq 4 describes how variance in spike phase results from the interaction between the

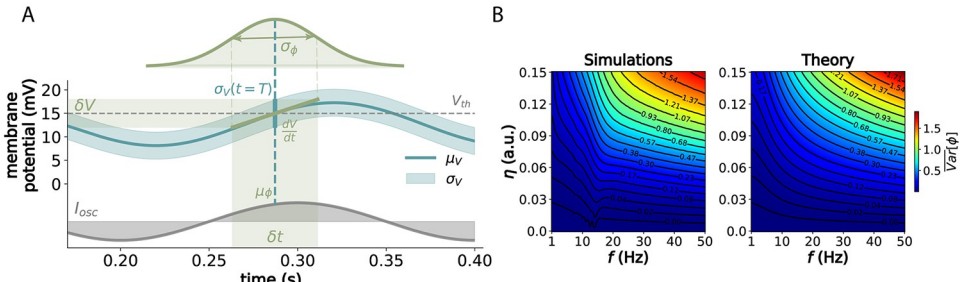

**Fig 2. Analytical approximation of phase distributions.** (A) Approximation of variance in phase of firing depends on the interplay between the linearized membrane dynamics at spike threshold $\frac{dV}{dt}|_{V_{th}}$ (with $\mu_V$ being the free membrane potential) and the accumulated noise in voltage $\sigma_V(t)$ at the expected firing time after one period $T$, so that $\delta V \equiv \sigma_V (t = T)$ and $\delta t \equiv \sigma_t$. The probability distribution on the upper part denotes how $\delta t$ translates to $\sigma_\phi$ with respect to the oscillation (lower part). For this example, we used the same parameters as in Fig 1, defined in Table A in S4 Appendix, for an $I_s$ of 90 pA. (B) Mean variance (in rad$^2$) across $I_s$ levels for physiologically-relevant parameters of frequency and noise, for both simulations and the analytical predictions from Eq 4.

accumulated random fluctuations in membrane potential and the membrane dynamics near the spike threshold. This linear approximation is valid for small deviations from the threshold (low noise) and a nearly linear response around it (suprathreshold regime). Notably, it agrees well with simulations (Fig 2B), predicting $\sigma_\phi^2$ accurately under physiologically-relevant parameters and noise levels ($\sigma_W = [0.01, 0.015]$ V$/\sqrt{s}$ in [6, 27], corresponding to $\eta = [0.1, 0.15]$ in Fig 2B). The predictions align with simulations until noise levels become unrealistically high ($\eta > 0.2$) and primarily at oscillation frequencies above 30 Hz (S2 Fig). These deviations are largely due to variance being unbounded in theory (Eq 4) but bounded to $\sim \pi^2/3$ rad in simulations (i.e., with phase becoming approximately uniformly distributed in $[0, 2\pi]$). Moreover, in the regime of very high variance, these differences are negligible considering the limited range of $[0, 2\pi]$, having a minimal impact on information estimates. Thus, we conclude that phase can be generally well characterized by a normal distribution with mean $\mu_\phi$ (Eq 3) and variance $\sigma_\phi^2$ (Eq 4).

## Approximation of information rate

Using the mean and variance of the phase distributions, we can approximate the neuron's information about the stimulus $I_s$ per cycle. Assuming $I_s$ is uniformly distributed across the phase-locking range (with $M$ equally-spaced $I_s$), the phase distribution forms a Gaussian mixture with $M$ means $\boldsymbol{\mu}$ and variances $\boldsymbol{\sigma}^2$. The mutual information between $I_s$ and $\phi$ can be approximated by considering an aggregate measure of the overall spread of the mixture (see S3 Appendix for details), obtaining:

$$I(I_s, \phi) \approx \frac{1}{2} \left( \log_2(\mathbb{E}[\boldsymbol{\sigma}^2] + \text{Var}[\boldsymbol{\mu}]) - \mathbb{E}\left[\log_2(\boldsymbol{\sigma}^2)\right] \right). \tag{5}$$

This equation provides a good approximation of the information a spike phase contains about $I_s$ at every cycle (Fig 3).

The efficient coding hypothesis [21] suggests that neurons maximize information transfer, minimizing redundancy. Viewing oscillations as rhythmic input sampling [15], we then cast the problem of encoding a signal $s$ in terms of maximizing information rate (bit/s). Simply multiplying information per cycle and oscillation frequency $f$ gives $r_{upper} \approx If$, an upper bound that assumes cycle independence. However, a more realistic approach needs to account for

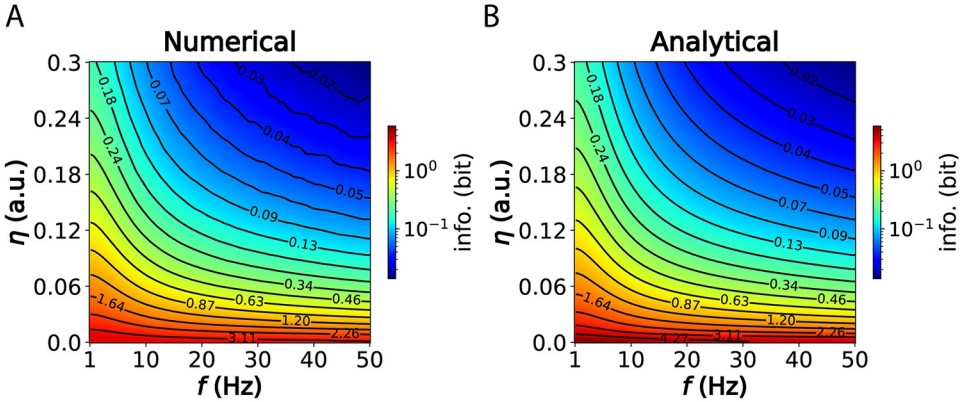

**Fig 3. Approximation of the mutual information.** (A) Mutual information via numerical integration of the Gaussian mixture (see S5 Appendix for details). (B) Analytical approximation using Eq 5. The color map is re-scaled to $\log_{10}$ for improved visualization.

cross-cycle correlations due to the relationship between the input signal autocorrelation $\rho(\Delta t) = e^{-\Delta t/\tau_s}$ (with characteristic time constant $\tau_s$) and the oscillation period $T$, hence modifying the information rate to (more details in S3 Appendix):

$$\mathrm{r} \approx \mathrm{r}_{upper}(1 - e^{-T/\tau_s}) = \mathrm{I}f(1 - e^{-T/\tau_s}). \tag{6}$$

This equation reflects the impact of temporal correlations, penalizing oversampling when $T < \tau_s$ by introducing an effective frequency factor (S3 Fig). Since we are mainly interested in finding the optimal frequency for given neuron parameters and noise level, we define the normalized information rate ($\mathrm{r}_{norm}$) as the ratio of r at a specific frequency ($\mathrm{r}_f$) to the maximum r across all frequencies ($F$):

$$\mathrm{r}_{norm} = \frac{\mathrm{r}_f}{\max_{f \in F}(\mathrm{r}_f)}. \tag{7}$$

Hence, this metric allows us to study the encoding capacity of neurons across the biologically relevant spectrum of oscillation frequencies.

## Results

### The speed-precision trade-off

Maximizing encoding precision requires operating at physiologically low frequencies to mitigate noise effects and enhance signal fidelity (Fig 4A). However, this approach inherently reduces the rate of input sampling, potentially resulting in delayed reaction times. In dynamic environments, where rapid responses are essential for survival, animals cannot afford to prioritize precision at the expense of speed. Conversely, operating at excessively high frequencies may lead to redundant input sampling and an amplification of noise (Fig 4A). Therefore, neural circuits must strike a balance that allows for both precise sensory information processing and sufficiently fast reaction times. This balance reflects a fundamental speed-precision trade-off faced by animals with inherently noisy brains (see [35] for a broader discussion).

The concept of information rate introduced earlier—defined as the expected number of bits transmitted per second—serves as a useful metric for capturing this trade-off, as it incorporates both the precision of encoded information and the speed of processing. In our theoretical

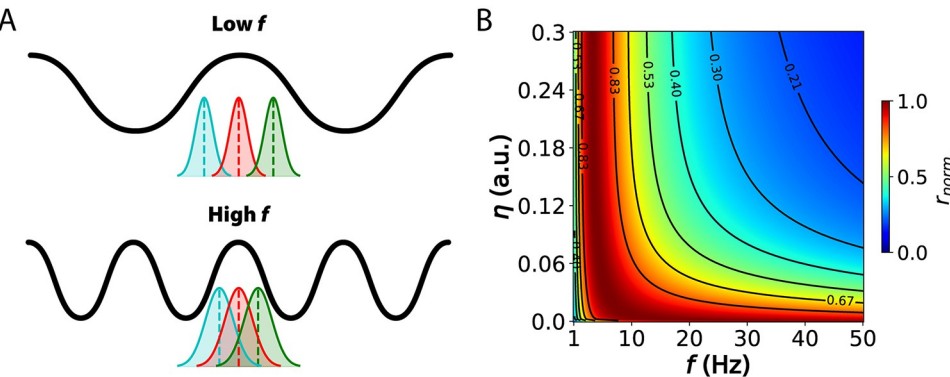

**Fig 4. Information rate is maximized in the theta band for hippocampal neurons.** (A) Implications for signal encoding at lower versus higher oscillation frequencies. (B) Information rate (bit/s, normalized across frequencies) for hippocampal neurons across a wide range of oscillation frequencies and noise strength levels. Neuron and oscillation parameters are detailed in Table A in S4 Appendix. The input characteristic time constant $\tau_s$ was set to 100 ms. Physiologically-realistic noise strength levels correspond to the range $\eta = [0.1, 0.15]$ [27].

model, we investigated this sampling-precision trade-off using physiologically realistic parameters of hippocampal neurons driven at various oscillatory frequencies and noise levels. Notably, we observed a peak in the information rate within the theta frequency range (3–8 Hz) across almost the entire noise spectrum (Fig 4B), corresponding to $\sim$1–2 bits/s for physiologically-realistic noise levels (S4 Fig). At very low noise levels, higher frequencies do not amplify noise sufficiently to significantly degrade information, thus remaining optimal. However, even a small increase in noise shifts the optimal frequency range to lower frequencies ($\sim$2–10 Hz). We further validated these theoretical findings with simulations (S5 Fig; see S5 Appendix for details). Additionally, we found that the optimality of the lower frequency range persists across a broad range of input signal time constants (S6 Fig) and membrane time constants (S7 Fig). This optimality also holds when pink noise is used instead of white noise (S9B Fig), lending further support to our results. Interestingly, we also identified a non-linear interaction between the membrane time constant and the input signal time constant that determines the optimality of the theta band (S8 Fig), revealing regimes within the parameter space where the theta band is suboptimal. Moreover, we demonstrate that physiologically unrealistic noise, such as brown noise, do not result in theta optimality across noise amplitudes (S9C Fig). Collectively, these additional results emphasize the specificity of theta optimality for realistic biological conditions.

## Theta optimality along the hippocampal dorsoventral axis

The hippocampus exhibits consistent variation in neuronal and circuit properties along its dorsoventral (DV) axis [36, 37]. Specifically, features such as membrane input resistance, action potential threshold, membrane time constant, and oscillation amplitude change nearly linearly along this axis [38, 39]. Notably, despite these gradients, theta frequency remains dominant across the DV axis [39] (Fig 5A). Given this observation and our previous results, we hypothesized that the co-variation of these physiological parameters ensures that theta optimizes the speed-accuracy trade-off across the entire DV axis. Using neuron models defined by the physiological gradients along the DV axis [38], we estimated the normalized information rate across oscillation frequencies. As predicted, theta frequency consistently peaked across the entire DV axis (see Fig 5B and S10 Fig for further validation by simulations), supporting its optimality for phase coding throughout the hippocampus.

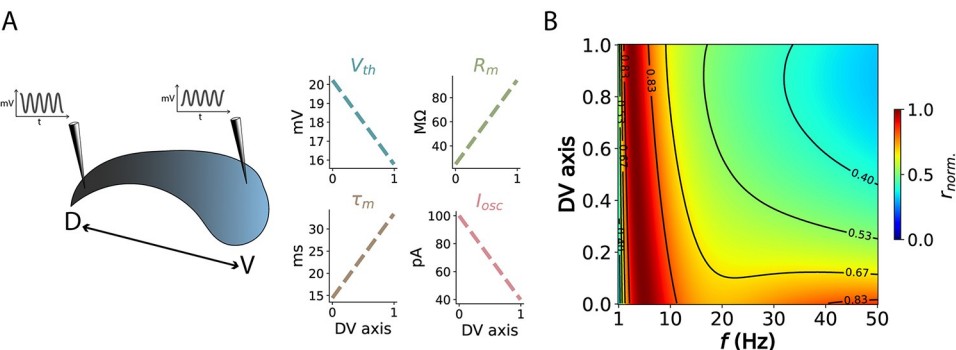

**Fig 5. Theta remains optimal along the dorsoventral axis.** (A) Physiological gradients along the DV axis of the hippocampus, extracted from [38] and [39]. (B) Information rate for neuron models along the DV axis, across frequencies. Detailed parameters can be found in Table B in S4 Appendix.

This theta optimality can be understood through the constancy of the $R_m/\tau_m$ ratio, which maintains a consistent transfer function (as shown in Figure 4E in [6]). A similar relationship can be deduced by recognizing the inverse relationship between these parameters in determining the spike phase variance in our Eq 4. Additionally, lower $I_{osc}$ in ventral regions are counterbalanced by increased excitability due to lower $V_{th}$ [40]. In conclusion, physiological gradients along the DV axis, likely reflecting different input processing needs [37, 41], co-vary to allow optimal input sampling at theta frequency throughout the hippocampus, thereby facilitating information flow and integration across regions.

## Optimal modulation of theta frequency and amplitude by locomotion speed

As animals move, they adjust their locomotion speed to meet task demands. Faster movement increases the rate of incoming stimuli, potentially requiring a higher frequency of perceptual sampling and corresponding brain rhythms (e.g., to normalize memory content [42]). Following this reasoning, and since theta oscillations govern the hippocampal processing of cortical inputs, it would be expected that theta frequency increases with speed (Fig 6A). Indeed, studies have shown a linear increase in theta frequency with locomotion speed in the rodent hippocampus, particularly in dorsal regions that receive sensory signals [43–45]. This relationship has also been observed in the human posterior hippocampus [46]. Furthermore, recent analyses indicate that it is specifically speed, not acceleration, that modulates both theta amplitude and frequency [47]. In turn, it has been suggested that hippocampal theta modulates local oscillatory circuits to maintain a consistent relationship between the rate of inputs from place field activity and spike phase [48]. Additionally, theta amplitude (and power) also exhibit a near-linear relationship with locomotion speed [43, 47, 49]. We then hypothesized that this concurrent modulation of theta frequency and amplitude with speed might reflect an underlying objective of maximizing information rate.

To test our hypothesis, we estimated the information rate profiles across different oscillation frequencies and amplitudes ($I_{osc}$). We observed a subtle yet consistent peak shift towards higher theta frequencies as the amplitude increased (Fig 6B and 6C; and S11 Fig for validation by simulations). This shift exhibited a nearly linear relationship before reaching saturation, consistent with previous findings. Therefore, our results suggest that maximizing information rate requires the concurrent modulation of both frequency and amplitude, providing an explanation for the observed correlation between oscillatory features and the animal's locomotion speed.

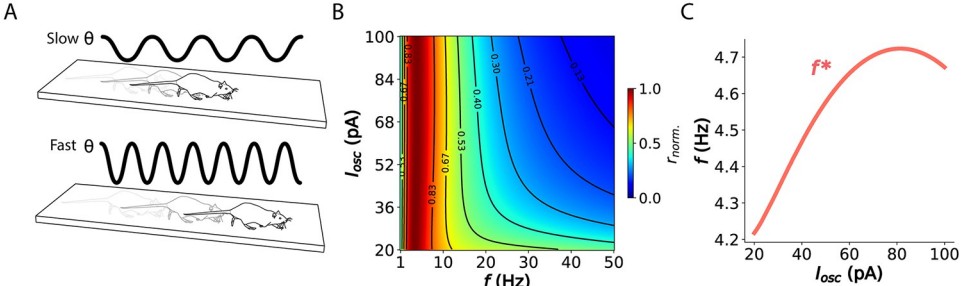

**Fig 6. Optimal sampling requires linear modulation of theta frequency and amplitude.** (A) Relationship between the rodent's locomotion speed on a linear track and hippocampal oscillation frequency. (B) Normalized information rate across frequencies and oscillation amplitudes. Neuron parameters can be found in Table A in S4 Appendix. (C) Optimal oscillation frequency defined as the information rate peak across oscillation amplitudes.

## Generalization to extra-hippocampal areas

Our results so far suggested that the information rate maximization principle might explain the presence and adaptation of theta oscillations in the hippocampus, as efficient phase-coding neurons overcome the trade-off between speed and precision during oscillatory input sampling. However, phase coding does not seem to be unique to the hippocampus. Pyramidal cells in the primary visual cortex of rodents [50–52] and monkeys [24, 53–55] also exhibit theta phase locking, which has been associated with feedforward processing of visual stimuli in superficial layers. Similarly, mitral cells in the olfactory bulb of mice are strongly modulated by theta oscillations that align with the respiratory cycle [9, 56, 57].

Based on these observations, we hypothesized that low-frequency oscillations would also maximize the information rate in these sensory areas. Consistent with our hypothesis, we found a peak in the high theta-low alpha band for both models: pyramidal neurons in primary visual cortex (Fig 7A) and mitral cells in the olfactory bulb (Fig 7B).

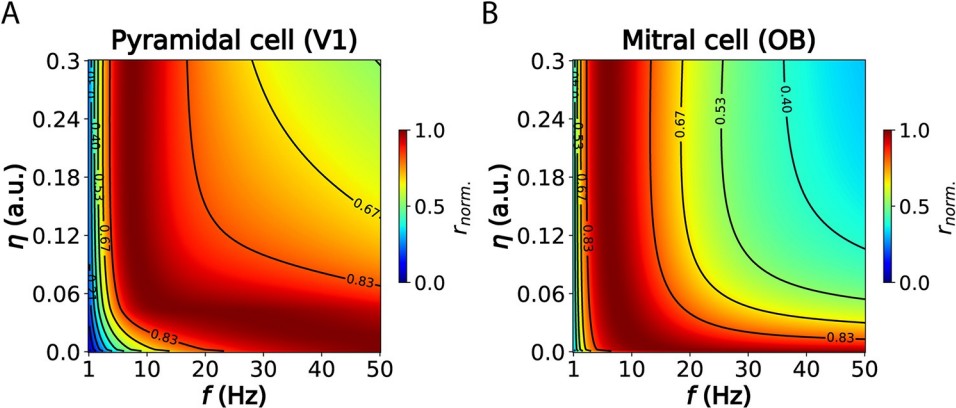

**Fig 7. Low frequencies maximize information rate in visual cortex and olfactory bulb.** (A) Normalized information rate across a broad frequency–noise parameter space for prototypical pyramidal cells in the primary visual cortex (V1) of mice [58]. A faster time constant of the input signal ($\tau_s$) of 50 ms was used to account for the rapid signal fluctuations in visual cortex. (B) Same as (A), but for mitral cells in the olfactory bulb of mice [9]. A slower $\tau_s$ of 250 ms was applied to match the cycle period of the respiratory rhythm. Detailed parameters of both neuron models can be found in Table C in S4 Appendix.

## Discussion

Our findings highlight the role of theta-band (3–8 Hz) oscillations in optimizing the encoding of information in phase-coding neurons, particularly within the hippocampus. By developing a theoretical framework that approximates the information rate conveyed by phase-coded signals, we have shown that theta oscillations represent an optimal solution to the trade-off between sampling speed and encoding precision. This trade-off is governed by the interplay between oscillation frequency and neuronal noise, resulting in a peak information rate within the theta frequency range.

### Relevant timescales and constraints in the hippocampus

Our results suggest that low-frequency, theta oscillations strike a balance that maximizes the information rate, enabling efficient phase coding that supports both precise sensory processing and rapid behavioral responses. Interestingly, our estimate of $\sim 1$–2 bits/s matches the overall slowness of behavioral output rate of $\sim 1$–10 bits/s reported in many different tasks and experimental paradigms [59]. In addition, the emergence of the theta range in the hippocampus as the optimal sampling frequency may impose constraints on the timescales of cortical input signals necessary for reliable encoding. Given that the bandwidth (B) of a signal with an exponentially-decaying autocorrelation function is $B = 1/(\pi \tau_s)$, and since according to the Nyquist-Shannon theorem the sampling frequency should satisfy $f \geq 2B$, we predict that cortical input signals (e.g., from the entorhinal cortex) should approximately follow $\tau_s \geq 80$ ms. Thus, signals with faster rates of change may risk communication losses, particularly if higher frequency ranges are employed to convey information. Therefore, we predict that rate-based input signals from the entorhinal cortex are characterized by relatively long time constants.

### Multi-modal, multi-scale integration through theta waves

Despite significant physiological gradients along the hippocampal dorsoventral (DV) axis, theta oscillations remain the dominant frequency across this axis. Our theoretical framework suggests that the co-variation of key physiological parameters optimizes the speed-precision trade-off, ensuring theta's persistence throughout the hippocampus. We suggest that this uniform theta frequency likely facilitates the integration and transfer of information across different spatial scales and levels of resolution [60], serving as well as a consistent readout mechanism [61]. In turn, dorsoventral traveling waves, which follow the DV axis [39], may further enable this phase-based integration. In addition, different regions along the DV axis process inputs with varying modalities and statistics, from slow olfactory signals to fast auditory ones. The theta-based phase code might then allow local tuning of single-cell properties to diverse inputs while integrating them within a common coding format, enabling standardized and multiplexed encoding.

### Speed-controlled oscillators for optimal sampling

The linear relationship between theta frequency-amplitude and locomotion speed in rodents supports our theoretical predictions, suggesting that the hippocampus adjusts these parameters to maintain optimal information rates and efficient sensory encoding. Indeed, theta frequency fluctuations with locomotion speed [43–45] are driven by increased cue densities, which may require higher frequencies to match stimulus rates and normalize information per theta cycle [42], thereby helping downstream circuits maintain consistent decoding [61]. However, we have shown that frequency increases alone may degrade information rates due to noise; therefore, concurrent amplitude increases are necessary to sustain optimal phase coding. The

medial septum, the primary driver of hippocampal theta oscillations [62–64], likely coordinates these adjustments, as it modulates both locomotion speed and theta frequency [65, 66]. Indeed, cooling experiments suggest the presence of a global oscillator in the medial septum that normalizes phase relationships by modulating theta frequency and amplitude [67]. We hypothesize that this oscillatory adaptive mechanism serves to stabilize neural processing and prevent disruptive synaptic changes during speed variations.

### Phase coding across brain regions and species

Our findings, though focused on the hippocampus, extend to other brain regions where phase coding has been observed, such as the primary visual cortex and olfactory bulb. We have shown that in these areas, low-frequency oscillations—particularly in the high theta to low alpha range—also maximize information rate. Therefore, our results suggest a broader applicability of the speed-precision trade-off framework and encourage exploration of how single-cell properties constrain optimal frequency and noise levels across different brain areas and species. For instance, although theta phase coding has been observed in humans [8, 23, 68], there appear to be two distinct theta rhythms in the human hippocampus [46], which seem to mirror the two types of hippocampal theta oscillations in rabbits and rats reported in early physiological studies [69]. These heterogeneity in theta rhythms might reflect different pyramidal neuron types with unique membrane properties tuned to process specific input characteristics. Furthermore, non-oscillatory phase coding in the hippocampus of bats [25] also raises questions such as whether rhythmic oscillations are needed to enable reliable phase coding. Lastly, despite the overall conservation of oscillations across mammalian species, the theta band seems to be the only one where the specific frequency decreases systematically with brain size [22]. This could be explained by considering that bigger animals tend to move more slowly than smaller ones, hence affecting the temporal correlations of sensory signals (i.e., the $\tau_s$ in our framework) and thus the optimal oscillation frequency. All in all, our framework offers a starting point to map single-cell properties to frequency regimes that optimize phase-coded information rate, assessing the plausibility of such coding schemes across brain circuits.

### Applications to neuromorphic computing and artificial neural networks

Recent advancements in training spiking neural networks akin to standard artificial neural networks [70, 71] have paved the way for optimizing large-scale spiking circuits for complex cognitive tasks, such as language production [72] and autonomous driving [73]. Incorporating efficient encoding strategies like phase coding, by using layer-wise reference oscillations or directly via complex-valued computations [74], could significantly enhance the large-scale training of deep spiking networks. These sparse encoding formats would also support energy-efficient neuromorphic chips, enabling inference and training at the edge [75]. Additionally, standard artificial neural networks—such as convolutional and recurrent neural networks—have been shown to benefit from reference oscillations in tasks related to visual perception [76] and working memory [77], suggesting promising directions for future research.

### Limitations and future work

Our theoretical framework relies on several assumptions that introduce limitations. One key assumption is the use of a noiseless model to estimate the expected spike phase, which does not account for the shift to earlier phases caused by high noise due to the asymmetry of spike threshold crossing. Additionally, by treating phase as a non-circular variable, we introduce a cutoff at $\phi = 0$ (i.e., oscillation trough), resulting in boundary effects in simulations, where strong inputs under high noise cluster near $\phi = 0$ (see S2A Fig, bottom-left panels, yellow

distributions). Furthermore, neurons can produce multiple spikes per cycle in simulations, leading to cycle-to-cycle interference at certain frequencies (e.g., see S2A Fig, 10 Hz column, yellow distribution). Moreover, although these extra spikes representing firing rate modulations are not captured by our model, evidence suggests that rate and phase modulations are largely independent in hippocampal neurons [8, 78]. Therefore, the implications of extra spikes during phase coding and the possible interactions between phase and rate remain unclear. Lastly, our estimation of phase variance is based on a linear approximation of system dynamics at threshold, which does not fully capture the underlying nonlinear dynamics. While this approximation is adequate for physiologically relevant noise levels (Fig 2B), it diverges from simulations at higher noise levels (see S2B and S2C Fig). Although this divergence is largely due to phase being bounded to the range of 0 to $2\pi$ in simulations but unbounded in theory, the abovementioned factors limit the applicability of our framework, possibly requiring numerical validation when medium-to-high noise levels are considered. Furthermore, our framework does not explicitly address other significant factors, such as metabolic cost (e.g., bit/spike efficiency), the possibility of population coding strategies (e.g., potentially reducing variance as $1/N$ via sample mean decoding), and their interactions: increasing neuron count would enhance precision but also raise metabolic costs, suggesting optimal population sizes. Therefore, future work should consider these aspects to better understand how they influence optimal sampling frequencies in phase coding neurons under realistic metabolic conditions.

## Supporting information

**S1 Appendix. Full derivation of the mean phase of firing.** Provides a detailed solution to the deterministic part of Eq 1, resulting in the "rate-to-phase" transfer function (Eq 3) previously derived in [6].
(PDF)

**S2 Appendix. Full derivation of the variance of phase of firing.** Explains the use of a first-order Taylor series expansion and the propagation of uncertainty around the spike threshold to derive an analytical approximation of the phase variance.
(PDF)

**S3 Appendix. Approximation of information rate.** Describes the approximation of the entropy of Gaussian mixtures to derive an analytical estimation of the information rate. Additionally, it introduces a correction factor to account for cycle-to-cycle correlations.
(PDF)

**S4 Appendix. Neuron parameters.** Includes: Table A. Default parameters for hippocampal neurons; Table B. Neuron parameters along the hippocampal dorsoventral axis; Table C. Neuron parameters for visual and olfactory cells.
(PDF)

**S5 Appendix. Simulations.** Details the numerical integration of Eq 1 used in simulations supporting our theoretical framework.
(PDF)

**S1 Fig. Oscillatory frequency modulates the tonic input range that makes neurons phase-lock.** (A) Phase-locking function for different values of the oscillatory frequency $f$ (color coded). Shaded areas denote the phase-locking range of $I_s$, corresponding to the domain of $I_s$ in Eq 3. The parameters are the same ones used in [6] and in Fig 1, to match hippocampal physiology (described in Table A in Appendix). Note that the phase-locking range spans half of a cycle and

not the full cycle as previously thought, in agreement with recent studies [7]. (B) Length of $I_s$ range $(\max(I_s) - \min(I_s) = 2AI_{osc})$ across frequencies. (C) Average $I_s$ as $\bar{I}_s = V_{th}/(R_m(1 - e^{-T/\tau_m}))$ (middle point in $I_s$ range) across frequencies. (D) Effective amplitude of the membrane potential oscillation, $V_{osc}$ produced by the oscillatory input $I_{osc}$. Given that the membrane acts as a low-pass filter, determined by $\tau_m$, the effective oscillation in the membrane potential can be found to be

$V_{osc} = R_m I_{osc} A$. Thus, since the membrane filters the oscillatory input as $A = 1/\sqrt{1 + (\tau_m 2\pi f)^2}$, the amplitude of the membrane potential $V_{osc}$ will decrease with $f$ approximately as $\sim 1/f$.
(TIF)

**S2 Fig. Analytical approximation of phase distributions.** (A) Phase distributions for a range of frequencies and noise strengths. Histograms denote the simulations whereas solid lines denote the theoretical predictions. For the simulations, first-spike phases are recorded from the beginning of the second cycle (with the trough as $\phi = 0$), after initializing the neurons to their expected phase $\phi_0 = \mu_\phi$ to allow them to reach steady-state dynamics (as described in Appendix). The parameters used here are described in Table A in Appendix). (B) Average variance in $\text{rad}^2$ (across $I_s$ levels) across a wide frequency–noise parameter space, for simulations and the theoretical predictions. (C) Diagonal slices of plots in (B), showing the deviation of the theory from the simulations after a certain level of noise amplitudes at high frequencies, due to the bounded variance of simulated spike phases constrained to the measurable range of $[0, 2\pi]$ radians.
(TIF)

**S3 Fig. Effective rhythmic input sampling.** (A) An example signal with $\tau_s$ of 100 ms sampled by different oscillation frequencies. (B) Effective frequency $f_{\text{eff}} = (1 - e^{T/\tau_s})f$ for various $\tau_s$ values.
(TIF)

**S4 Fig. Information rate across frequencies for the range of physiologically realistic noise levels ($\eta$).**
(TIF)

**S5 Fig. Normalized information rate across the frequency–noise parameter space for simulations and theoretical predictions.**
(TIF)

**S6 Fig. Normalized information rate across the frequency–noise parameter space for a wide range of input signal time constants $\tau_s$.**
(TIF)

**S7 Fig. Normalized information rate across the frequency–noise parameter space for a wide range of membrane time constants $\tau_m$.**
(TIF)

**S8 Fig. Optimal frequency for a wide range of membrane time constants $\tau_m$ and input signal time constants $\tau_s$.** At every point of the $\tau_m - \tau_s$ parameter space (logarithmically discretized in a $200 \times 200$ grid), we computed $r_{norm}$ over the frequency–noise space (as in e.g., Fig 4B). Then, the optimal frequency was estimated as an average of the peak frequency between the physiologically-realistic noise range $\eta = [0.1, 0.15]$.
(TIF)

**S9 Fig. Normalized information rate for colored noise with different long-range correlation lengths: White, pink, and brown noise.** A value of 100 ms was used here for $\tau_s$. All plots

represent the results of simulations.
(TIF)

**S10 Fig. Normalized information rate across the dorsoventral axis for simulations and the-oretical predictions.**
(TIF)

**S11 Fig. Normalized information rate across the frequency-amplitude space for simula-tions and theoretical predictions.**
(TIF)

## Acknowledgments

We would like to thank Francisco Páscoa dos Santos, Raimon Bullich Vilarrubias, Ismael Tito Freire, Óscar Guerrero Rosado, and the entire SPECS-lab team for their invaluable feedback throughout the development of this manuscript. We are also deeply grateful for the insightful discussions with Daniel Pacheco Estefan, Diogo Santos Pata, and Francky Catthoor during the earlier stages of this work. Additionally, we thank Bernat Molero Agudo for his efforts in test-ing the earliest versions of the model.

## Author Contributions

**Conceptualization:** Adrián F. Amil, Albert Albesa-González, Paul F. M. J. Verschure.

**Formal analysis:** Adrián F. Amil, Albert Albesa-González.

**Funding acquisition:** Paul F. M. J. Verschure.

**Software:** Adrián F. Amil, Albert Albesa-González.

**Supervision:** Paul F. M. J. Verschure.

**Writing – original draft:** Adrián F. Amil, Albert Albesa-González.

**Writing – review & editing:** Adrián F. Amil, Albert Albesa-González, Paul F. M. J. Verschure.

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
