## [Decision Letter · Decision Letter 0]

12 Nov 2024

Dear Mr. Amil,

We are pleased to inform you that your manuscript 'Theta oscillations optimize a speed-precision trade-off in phase coding neurons' has been provisionally accepted for publication in PLOS Computational Biology.

Best regards,

Daniel Bush

Academic Editor

PLOS Computational Biology

Marieke van Vugt

Section Editor

PLOS Computational Biology

Feilim Mac Gabhann

Editor-in-Chief

PLOS Computational Biology

Jason Papin

Editor-in-Chief

PLOS Computational Biology

Reviewer's Responses to Questions

**Comments to the Authors:**

Reviewer #1: The Authors reworked the manuscript substantially. They addressed all listed concerns. The text states model constraints much clearer and the added results and figures much better support the author's claims.

I am satisfied with the revised manuscript an recommend its publication.

Reviewer #2: In general, this is a fairly interesting topic, whether or not the firing rate properties/noise filtering/noise levels present for cells will impact the frequency of the theta oscillation, i.e. the sampling frequency. I think the authors have addressed the criticisms of the previous reviewers effectively here, with additional simulations, analytical work, and rewriting for clarity/additional references. The analysis is thorough, although not necessarily novel (Reviewer 1s comments), but the application, to my knowledge, is novel. The modelling decisions are also well informed by more recent research (e.g. just using the first spike). There's probably additional extensions that can be considered like considering more spikes in a place field, or the impacts of non-integrator (e.g. resonator) models, impacts of spike frequency adaptation, etc., but these are better followed up in future work.

Minor comment

I'm finding most of the colour maps to have unreadable text (level set heights) when the density of lines is set too high (e.g., Figure 4, near the 1Hz line.

**Have the authors made all data and (if applicable) computational code underlying the findings in their manuscript fully available?**

Reviewer #1: Yes

Reviewer #2: Yes

PLOS authors have the option to publish the peer review history of their article (what does this mean?). If published, this will include your full peer review and any attached files.

Reviewer #1: No

Reviewer #2: No

---

## [Editor Report · Acceptance letter]

25 Nov 2024

PCOMPBIOL-D-24-01713 

Theta oscillations optimize a speed-precision trade-off in phase coding neurons

Dear Dr Amil,

I am pleased to inform you that your manuscript has been formally accepted for publication in PLOS Computational Biology. Your manuscript is now with our production department and you will be notified of the publication date in due course.

With kind regards,

Olena Szabo
